# Chemical Composition, Phytotoxic and Antibiofilm Activity of Seven Eucalyptus Species from Tunisia

**DOI:** 10.3390/molecules27238227

**Published:** 2022-11-25

**Authors:** Sana Khedhri, Flavio Polito, Lucia Caputo, Francesco Manna, Marwa Khammassi, Lamia Hamrouni, Ismail Amri, Filomena Nazzaro, Vincenzo De Feo, Florinda Fratianni

**Affiliations:** 1Faculty of Science, Bizerte, Zarzouna 7021, Tunisia; 2Department of Pharmacy, University of Salerno, 84084 Fisciano, Italy; 3Laboratory of Management and Valorization of Forest Resources, National Institute of Researches on Rural Engineering, Water and Forests, P.B. 10, Ariana 2080, Tunisia; 4Laboratory of Biotechnology and Nuclear Technology, National Center of Nuclear Sana Science and Technology, Sidi Thabet, B.P. 72, Ariana 2020, Tunisia; 5Insitute of Food Science, CNR-ISA, 83100 Avellino, Italy

**Keywords:** essential oil, *Eucalyptus*, phytotoxic activity, antibiofilm activity

## Abstract

This study was carried out to characterize the chemical composition of the essential oils from seven *Eucalyptus* species (*E. griffithsii*, *E. hemiphloia*, *E. lesouefii*, *E. longicornis*, *E. pyriformis*, *E. viminalis*, and *E. wandoo*), as well as their phytotoxic and antibacterial activities. The essential oils were analyzed by GC/MS and the potential in vitro phytotoxicity was evaluated against germination and radical elongation of *Raphanus sativus*, *Lolium multiflorum*, and *Sinapis arvensis* seeds. The antibiofilm activity was studied against both Gram-negative (*Pseudomonas aeruginosa, Escherichia coli* and *Acinetobacter baumannii*) and Gram-positive (*Staphylococcus aureus* and *Listeria monocytogenes*) bacteria. The inhibition of biofilm formation and its metabolism was determined at different times. Eucalyptol was the most abundant component in all essential oils studied (ranging from 40.8% for *E. lesouefii* EO to 73.6% for *E. wandoo*) except for that of *E. pyriformis* where it was present but at 15.1%. *E. pyriformis* was the most active against both germination and radical elongation of *S. arvensis*. The action of all essential oils proved to be highly effective in inhibiting the bacterial adhesion process of the five strains considered. In light of these results, these essential oils could have potential applications both in the agricultural and health fields.

## 1. Introduction

The *Eucalyptus* genus (Myrtaceae) is composed of evergreen aromatic flowering trees and comprises about 800 species. It is a native to Australia although *Eucalyptus* species are now distributed in almost all parts of the world [1].

Different parts of the plant have been used to produce essential oils (EOs), but in the leaves these oils were most plentiful [2]. The EOs from *Eucalyptus* species rich in eucalyptol have traditionally been utilized as disinfectants, insect repellents, febrifuges and to treat respiratory illnesses [3]. Nowadays, they are used in many pharmaceutical and cosmetics industries [2].

Moreover, the identification of bioactive allelochemicals in several *Eucalyptus* species EOs developed research towards potential natural herbicidal compounds for weed management [4,5,6,7]. Nowadays, only 8% of conventional herbicides derive from natural sources even though natural product-based herbicides are considered safer than conventional agrochemicals [8]. However, beyond human and environmental toxicity, many weed plants are developing resistance to classical synthetic herbicide; therefore, it appears necessary to increase the study of new natural herbicidal substances [9].

On the other hand, biofilm formation and drug resistance found in both Gram-negative and Gram-positive bacteria represent problems for human health [10,11]. Several studies have demonstrated the antimicrobial properties of the EOs from *Eucalyptus* species [12,13] but few studies investigated their anti-biofilm activity [14].

The present study was carried out to characterize the chemical composition of the EOs from seven *Eucalyptus* species (*E. griffithsii* Maiden, *E. hemiphloia* Benth. (*E. moluccana* Wall. ex Roxb.), *E. lesouefii* Maiden, *E. longicornis* (F. Muell.) Maiden, *E. pyriformis* Turcz., *E. viminalis* Labill. and *E. wandoo* Blakely), as well as their phytotoxic and antibiofilm activities. The potential in vitro phytotoxicity was evaluated against germination and radical elongation of *Raphanus sativus* L. (radish), *Lolium multiflorum* Lam. (Italian ryegrass), and *Sinapis arvensis* L. (wild mustard) seeds, whereas the antibiofilm activity was evaluated against two Gram-positive (*S. aureus* and *L. monocytogenes*) and Gram-negative (*E. coli*, *P. aeruginosa* and *A. baumannii*) pathogenic strains.

## 2. Results and Discussion

### 2.1. Chemical Composition

The EO yields from leaves of the *Eucalyptus* species are shown in Table 1, ranging from 1.3 (*E. wandoo*) to 3.6% (*E. hemiploia*). The available literature reports yields of 5.2 and 2.0%, respectively for the EOs from leaves of Tunisian *E. lesouefii* and *E. wandoo* [15]. The yields of *E. griffithsii*, *E. hemiphloia*, *E. pyriformis*, *E. viminalis* were higher than those reported for *E. griffithsii* from Australia (0.34%) [16], *E. hemiploia* from Morocco (1.24%) [17], *E. pyriformis* from Australia (1.53%) [18], and *E. viminalis* from Portugal (1.1%) [19]. No studies reported the EO yield of *E. lesouefii*.

The percent compositions of the EOs are reported in Table 2. The compounds are listed according to their elution order on a HP-5MS column. The oxygenated monoterpenes were the main constituents in all EOs, with a percentage ranging from 53.9 (*E. pyriformis*) to 92.8% (*E. longicornis*). The highest amount of hydrocarbon monoterpenes was found in *E. pyriformis* EO (36.2%), while sesquiterpene hydrocarbons were present in very low quantity and only in *E. lesouefii* (0.1%), *E. pyriformis* (0.7%), *E. viminalis* (1.2%) and *E. wandoo* (0.9%) EOs. Oxygenated sesquiterpenes were most abundant in *E. wandoo* EO (5.8%).

Eucalyptol was the most abundant component in all EOs, ranging from 40.8 (*E. lesouefii*) to 73.6% (*E. wandoo*), except for *E. pyriformis* EO, where it is present at 15.1%. 

Forty three components were identified in the EO of *E. griffithsii*, accounting for 96.7% of the total oil. Eucalyptol (48.9%), α-pinene (9.8%), *o*-cymene (4.2%) and *trans*-pinocarveol (4.6%) were the main components. These data agree with a previous analysis of EO from *E. griffithsii* from Iraq, where eucalyptol and α-pinene were reported in lower amounts (26.1 and 7.9%, respectively) and *trans*-pinocarveol in a higher quantity (10.6%) with respect to our sample [16]. Other compounds, in amounts greater than 1%, were *neo*-verbenol (2.9%), α-terpineol (1.9%), cumin aldehyde (1.9%), terpinene-4-ol (1.8%), dihydrocarveol (1.6%), pinocarvone (1.3%), *p*-menth-1-en-7-al (1.2%) and carvacrol (1.1%). 

In the EO of *E. hemiphloia*, 35 components were identified, accounting for 97.5% of the total EO. Eucalyptol (63.2%), *p*-cymene (5.5%) and *iso*-menthol (4.6%) were the main compounds. Zrira and coworkers reported eucalyptol as the principal constituent (44.2%) of a Moroccan EO, but the other main components were α-pinene (7.8%), β-pinene (5.8%), α-terpineol (7.0%) that in our EO were present in smaller amounts or totally absent [17].

The EO from *E. lesouefii* showed the presence of 24 components, accounting for 96.8% of the total EO. Eucalyptol (40.8%), α-pinene (14.6%), *p*-cymene (14.3%), *trans*-pinocarveol (4.9%), β-pinene (4.4%) were the principal components. These results agree in part with Ameur and coworkers that reported eucalyptol (38.0%), α-pinene (12.8%), *p*-cymene (7.7%), *trans*-pinocarveol (3.2%), β-pinene (10.9%) and spathulenol (4.6%) as the main components of an EO from the North West of Tunisia [15].

Twenty-one compounds were identified in the EO of *E. longicornis*, accounting for 97.6% of the total EO, with eucalyptol (84.2%), α-pinene (3.5%), and *trans*-pinocarveol (4.2%) as the main components. To the best of our knowledge, the chemical composition of the EO of *E. longicornis* has not yet been reported.

In the EO of *E. pyriformis*, 46 components were identified, accounting for 97.3% of the total EO, in which the main constituent was *p*-cymene (28.8%). eucalyptol (15.1%), *m*-cymen-8-ol (8.8%), *trans*-pinocarveol (4.4%), sabina ketone (3.9%), β-pinene (3.5%) and α-pinene (1.9%) were the other principal components. Instead, Bignell and coworkers reported eucalyptol as the principal constituent (39.2%) of an EO from South Australia, with aromadendrene (5.2%), β-eudesmol (4.7%), limonene (2.4%), *trans*-pinocarveol (1.7%) and α- eudesmol (1.7%) [18]. Among these compounds, aromadendrene, limonene and β-eudesmol were totally absent in our sample.

The EO from *E. viminalis* showed the presence of 31 components, accounting for 96.6% of the total EO. Eucalyptol (68.1%), α-pinene (7.2%), *trans*-pinocarveol (3.9%), spathulenol (5.1%) were the principal components. Other compounds present in lesser amounts were sabina ketone (1.2%), pinocarvone (1.2%), terpinen-4-ol (1.1%) and cryptone (1.4%). These results agree in part with Elaissi and coworkers who showed in an EO from the North of Tunisia the presence of eucalyptol (62.7%), α-pinene (1.7%), *trans*-pinocarveol (2.3%) and *p*-cymene (1.3%), the last absent in our sample, and the presence of some oxygenated sesquiterpenes (globulol, viridiflorol and β-eudesmol), totally absent in our sample [20]. Moreover, an Iranian EO showed eucalyptol (57.8%), α-pinene (13.4%), limonene (5.4%), and globulol (3.0%) as the main constituents [21]. Limonene and globulol were absent in our EO. 

In the EO of *E. wandoo*, 25 components were identified, accounting for 98.0% of the total EO, with eucalyptol (73.6%), *trans*-pinocarveol (6.1%), α-pinene (4.3%) as the principal constituents. Other components in lesser amounts were α-terpinyl acetate (1.9%), pinocarvone (1.3%) and the sesquiterpenes, spathulenol (2.1%), β-eudesmol (1.5%), rosifoliol (1.0%) and α-eudesmol (0.9%). The literature reports an Algerian EO much poorer in eucalyptol and *p*-cymene (14.9 and 9.0%, respectively), but with a high content of benzaldehyde (32.3%) absent in the sample analyzed [22]. Instead, a Tunisian EO showed a qualitative composition similar to that of our EO, but with different percentages of the individual constituents [15].

In general, the research agrees with the available literature; however, some different compositions have been found, in particular with regard to the composition of the EO of *E. pyriformis*. It is interesting to note that both genetic and environmental factors can influence the composition of the essential oils of species grown outside their original area.

### 2.2. Phytotoxic Activity

This is the first manuscript that investigates the phytotoxicity of these *Eucalyptus* EOs against germination and radical elongation of *S. arvensis*, *R. sativus* and *L. multiflorum* seeds. Figure 1, Figure 2, Figure 3, Figure 4, Figure 5 and Figure 6 represented the number of germinated seeds and the radicle length in cm of these seeds after treatment with the EOs. 

The germination and radical elongation of *R. sativus* seeds were significantly inhibited by all doses of all essential oils tested, except for *E. wandoo* EO. In fact, this last EO never inhibited germination or radical elongation to a percentage greater than 50%. 

The EOs from *E. lesouefii* and *E. pyriformis* were the most active against germination and radical elongation of *S. arvensis*. In fact, at the lowest doses tested (125 µg/mL), the inhibition of germination was 93.1 and 96.6%, respectively, and the inhibition of radical elongation was 90.6 and 94.8%, respectively. *E. griffithsii* and *E. viminalis* EOs were the second most active against *S. arvensis* germination with an inhibition > 50% at 125 µg/mL. Instead, the others EOs, at the lowest dose tested, showed a low inhibition of *S. arvensis* germination ranging from 3.3 to 43.4%. Moreover, the second most active EOs on radical elongation of *S. arvensis* were *E. griffithsii*, *E. hemiphloia* and *E. viminalis*. In fact, at 125 µg/mL, these EOs showed an inhibition on radical growth of 67.6%, 64.1% and 70.4%, respectively. *E. longicornis* and *E. wandoo* EOs were the less active, also against radical elongation of *S. arvensis* seeds. 

*E. hemiphloia*, *E. lesouefii, E. pyriformis*, *E. viminalis* EOs totally inhibited the germination and radical elongation of *L. multiflorum*, at the highest dose tested. *E. lesouefii* EO was the most active against *L. multiflorum* germination with a percentage inhibition of 63% at 500 µg/mL. Instead, *E. pyriformis* EO showed a better activity against *L. multiflorum* radical growth with a percentage inhibition of 82.7% at 250 µg/mL.

To the best of our knowledge, the phytotoxic activity on *S. arvensis*, *R. sativus* and *L. multiflorum* seeds of these *Eucalyptus* species has not been reported. 

Many authors reported the phytotoxic and allelopathic effects of several *Eucalyptus* EOs such as *E. citriodora* Hook, *E. nicholii* Maiden and Blakely, *E. globulus* Labill. and *E. tereticornis* Sm. against germination and growth of many crops and weeds [5,6,24,25].

The phytotoxicity was probably due to the presence of eucalyptol: as suggested in the literature this compound can causes decreased germination, inhibiting mitochondrial respiration, mitosis and DNA synthesis [4,26].

However, the phytotoxicity of the EO of *E. pyrifiormis* can be attributed to other constituents or to a synergism between them.

Our results appear relevant; in fact, all EOs affected germination and radical elongation of *S. arvensis,* one of the most prevalent weed species in wheat in the Middle Black Sea Region of Turkey [27]. Moreover, the EOs were active against *L. multiflorum,* one of the weeds that are evolving resistance to herbicides [28].

The results obtained may open the way for the direct application of essential oils studied in sustainable agricultural practices or in the identification of new lead compounds for new agrochemicals. 

### 2.3. Antibacterial Activity

Table 3 shows the minimal inhibitory concentration of the *Eucalyptus* EOs necessary to inhibit the bacterial growth of the five pathogens used in our study.

These MIC values helped us to appraise the aptitude of the seven EOs to affect the bacterial biofilm (Table 4) and to alter the metabolism of the sessile bacterial cells (Table 5).

As reported in Table 3, the MIC values varied between 25 and 38 µL/mL Based on such results, two sub-lethal doses (10 and 20 µL/mL) were used in the antibiofilm tests.

The action of all EOs proved to be highly effective in inhibiting the bacterial adhesion process, prodrome to the biofilm constitution of the five pathogens used in our experiments. The inhibition percentages at the highest concentration (20 µL/mL), were never lower than 58% (EV vs. *S. aureus*). The EOs resulted in an inhibition even up to 95.04% (EP vs. *S. aureus*), and in many cases the inhibition was higher than 80%. Overall, all the strains were sensitive to the action of the EOs. It is interesting to note that in some cases, when the EOs were added 24 h after the beginning of the bacterial growth process, when the bacteria were allowed to consolidate the biofilm on the walls of the multiwells, the inhibitory effectiveness of the EOs remained practically intact. Indeed, with 10 µL/mL of the EO of *E viminalis*, the inhibitory efficacy increased by more than ten times, from 4.36 to 50.77%. Furthermore, the EO of *E. wandoo* was more effective against *S. aureus* than in the adhesion event. At the same concentration, the inhibitory efficacy of this EO was practically doubled, from 59.16 to 94.55%. 

The EOs were active inhibitors of the sessile cell metabolism, at the early stage of bacterial biofilm formation, and the inhibitory activity on adhesion processes was mainly caused by the action exerted on cell metabolism. It is the case, for example, of *A. baumannii*, against which, with a few exceptions, the inhibition rates resulted in no less than 53.64% (*E. viminalis*), and which reached more than 83% (*E. griffithsii*, *E. hemiphloia*, and *E. pyriformis*). In the case of *E. coli*, the higher concentration of EOs used, provoked an inhibition as high as 90.7% (*E. lesoufii*) and never less than 63.66% (*E. wandoo*). In general, however, all EOs showed apparent inhibitory efficacy on the cellular metabolism of all strains. Such efficacy was also observed in tests performed on mature biofilms. Although it decreased in some cases (e.g., *E. griffithsii* vs. *E. coli* and *L. monocytogenes*, or *E. hemiphloia* and *E. viminalis* vs. *A. baumannii*, *E. coli*, *L. monocytogenes*, and *P. aeruginosa*), the strength of action of the EOs on sessile cell metabolism in mature biofilms was maintained. This shows that the inhibitory activity of these EOs was primarily explicated by acting on cell metabolism. The low inhibitory activity recorded in the MTT test, but high in the crystal violet test, may mean that the EOs probably acted by other inhibitory mechanisms (e.g., on bacterial cell walls, nucleic acids), as widely demonstrated for *Eucalyptus* EOs [29,30] and other phytochemicals [31]. The EOs we analyzed exhibited high inhibitory activity, demonstrating that they could act both in the initial bacterial cell adhesion events and on mature biofilms. This again shows the extreme versatility of these EOs against Gram-positive and Gram-negative pathogens. Luìs and coworkers showed the antibacterial and anti-quorum sensing activity of the EOs of *E. globulus* Labill. and *E. radiata* A. Cunn. ex DC. against *A. baumannii*, thus indicating how these EOs can block the processes leading to bacterial biofilm adhesion and formation from the very beginning [11]. *E. globulus* EO has been shown to act on the biofilm of *E. coli*, resulting in 62% inhibition [32], and *P. aeruginosa* [33]. The inhibitory efficacy of the studied EOs may help to expand the versatility in the use of *Eucalyptus* EOs. The EO of *E. globulus* counteracted the formation of biofilms of cariogenic bacteria [34], including *S. aureus*, against which the EOs we analyzed showed an effective inhibitory action not only in blocking the biofilm from the outset but, where it was already established, acting with an inhibitory vigor that reached up to 95%. Future works will study, at the molecular level, the genes involved in the bacterial adhesion and how, and if, there is a change from the immature biofilm to its mature form.

## 3. Material and Methods

### 3.1. Plant Materials

Leaves of the seven *Eucalyptus* species were harvested from different Tunisian arboretums located in different regions (Table 6). For each species, five samples harvested from more than five different trees were collected and mixed for homogenization. The leaves were stored in a dry place for fifteen days. Specimens were identified at the Regional Station of the National *research institute of rural engineering*, Waters and Forests (INRGREF).

### 3.2. Extraction

One hundred grams of dried leaves were submitted to water distillation (500 mL of water) for 4 h, using a Clevenger-type apparatus. The EOs were solubilized in *n*-hexane, dried in a N_2_ atmosphere, and stored in amber vials in the refrigerator at 4 °C.

### 3.3. Analysis of the Essential Oils

Analytical gas chromatography (GC) was carried out on a Perkin-Elmer Sigma-115 gas chromatograph (Perkin Elmer, Waltham, MA, USA) equipped with a flame ionization detector (FID) and a data handling processor. The separation was achieved using a HP-5 MS fused-silica capillary column (30 m × 0.25 mm i.d., 0.25 μm film thickness, Agilent, Roma, Italy). Column temperature: 40 °C, with 5 min initial hold, and then to 270 °C at 2 °C/min, 270 °C (20 min); injection mode, splitless (1 μL of a 1:1000 *n*-hexane solution).; injector and detector temperatures were 250 °C and 290 °C, respectively. The analysis was also run using a fused silica HP Innowax polyethylene glycol capillary column (50 m × 0.20 mm i.d., 0.25 μm film thickness, Agilent, Roma, Italy). In both cases, helium was used as the carrier gas (1.0 mL/min). GC/MS analyses were performed on an Agilent 6850 Ser. II apparatus (Agilent, Roma, Italy), fitted with a fused silica DB-5 capillary column (30 m × 0.25 mm i.d., 0.33 μm film thickness, Agilent, Roma, Italy), coupled to an Agilent Mass Selective Detector MSD 5973; ionization energy voltage 70 eV; electron multiplier voltage energy 2000 V. Mass spectra (MS) were scanned in the range 40–500 amu, scan time 5 scans/s. Gas chromatographic conditions were as reported in the previous paragraph; transfer line temperature, 295 °C. Most constituents were identified by GC by comparison of their Kovats retention indices (Ri), determined relative to the retention times (tR) of *n*-alkanes (C_10_–C_35_), with either those of the literature [35,36,37] and mass spectra on both columns or those of authentic compounds available in our laboratories by means of NIST 02 and Wiley 275 libraries [38]. The component relative concentrations were obtained by peak area normalization. No response factors were calculated.

### 3.4. Phytotoxic Activity

The phytotoxic activity was evaluated on germination and radical elongation of the seeds of *Lolium multiflorum* Lam., *Raphanus sativus* L., and *Sinapis arvensis* L., often used for their easy and well-known germinability. *S. arvensis* seeds were collected from wild populations in Sidi Ismail, Beja, Tunisia on June 2021. *R. sativus* seeds were purchased from Blumen group s.r.l. (Bologna, Italy) whereas *L. multiflorum* seeds were purchased from Fratelli Ingegnoli s.p.a. (Milano, Italy). The seeds were sterilized in 95% ethanol for 15 s and sown in Petri dishes (Ø = 90 mm), on three layers of Whatman filter paper, impregnated with 7 mL of deionized water used as first control to verify the germinability of the seeds, 7 mL of a water–acetone mixture (99.5:0.5, *v*/*v*) as second control (because the EOs were dissolved in this mixture for their lipophilicity) or 7 mL of different doses (1000, 500, 250, and 100 mg/mL) of the EOs. Controls, carried out with the water–acetone mixture alone, showed no differences in comparison with control in water alone. The germination conditions were 20 ± 1 °C, with a natural photoperiod. Seed germination was observed in the Petri dishes every 24 h. A seed was considered germinated when the protrusion of the root became evident [39]. On the fifth day (after 120 h) the effects on radicle elongation were measured in cm. Each determination was repeated three times, using Petri dishes containing 10 seeds each. Data are expressed as the mean ± SD for both germination and radicle elongation.

### 3.5. Antimicrobial Activity

#### 3.5.1. Microorganisms and Culture Conditions

*Acinetobacter baumannii* ATCC 19606, *Pseudomonas aeruginosa* DSM 50,071 and *Escherichia coli* DSM 8579 (Gram-negative strains), *Staphylococcus aureus* subsp. *Aureus* Rosebach ATCC 25923, and *Listeria monocytogenes* ATCC 7644 (Gram-positive strains) were used. Bacteria were grown in Luria broth for 18 h at 37 °C before the analysis. *A. baumannii* was cultured at 35 °C under the same conditions.

#### 3.5.2. Minimal Inhibitory Concentration (MIC)

The essential oils and DMSO were sterilized by ultrafiltration before use. The method using resazurin, developed by Sarker and coworkers [40], with some modifications was used to assess the MIC of the EOs in 96 microtiter-plates. Resazurin is an oxidation–reduction indicator, generally used to evaluate cell growth. It is a blue non-fluorescent and non-toxic dye, which becomes pink and fluorescent when is subjected to a reduction through the action of the oxidoreductases present within viable cells and is transformed to resorufin. This last molecule is further reduced to hydroresorufin that appears (uncolored and non-fluorescent). A resazurin solution was prepared by dissolving 270 mg in 40 mL of sterilized deionized water. One hundred μL of samples in DMSO (1:10 *v*/*v*) were added to the first row of the plate. To all other wells, 50 μL of Luria–Bertani broth or normal sterile solution was included. Serial descending concentrations of samples were performed, and each well was supplemented with 10 μL of resazurin indicator solution. Thirty μL of 3.3× strength isosensitized broth and 10 μL of bacterial suspension (5 × 10^6^ cfu/mL) were added in each well. The plates were then closed with parafilm to prevent dehydration. The wide-spectrum conventional antibiotic tetracycline, previously suspended in DMSO, and used in other similar research [41,42] was added in a column of the plate as positive control. Luria–Bertani broth containing resazurin and bacteria, without samples, was considered as the negative control. The plates were incubated at 37 °C (or at 35 °C for *A. baumannii*) for 24 h. The color change was then assessed visually. Any color changes from purple to pink or colorless were recorded as positive. The lowest concentration of EOs which could prevent the solution from changing from dark purple to pink was considered as the MIC value. 

#### 3.5.3. Biofilm Inhibitory Activity

The inhibitory activity of the EOs on the bacterial adhesion was assessed using flat-bottomed 96-well microtiter plates [42]. Bacterial cultures were adjusted to 0.5 McFarland with fresh culture broth. Then, 10 µL of the bacterial cultures and 10 or 20 µL/mL of the EOs were included in each well. The volume of each well was brought to 250 µL final volume with different amounts of Luria–Bertani broth. The plates were covered with parafilm tape to prevent evaporation and incubated for 48 h at 37 °C (or at 35 °C for *A. baumannii*). Succeeding the elimination of the planktonic cells, sessile cells were washed twice with sterile PBS. After PBS removal, the plates were left under a laminar flow hood for ten minutes. Two hundred µL of methanol was included in each well, to fix the sessile cells, and eliminated after 15 min. The plates were left to dry the samples and 200 µL of 2% *w*/*v* crystal violet solution/well was used for 20 min to stain the sessile cells. After 20 min, the staining solution was removed; plates were gently washed with sterile PBS and samples dried. The bound dye’s release was obtained by adding 200 µL of glacial acetic acid 20% *w*/*v*. The absorbance was measured at λ = 540 nm (Cary Varian, Milano, Italy). The percent value of adhesion was assessed with respect to the control (formed by the cells grown without the presence of the samples, where a value of 0% inhibition was considered). Triplicate tests were performed, and the average results were calculated for reproducibility.

#### 3.5.4. Activity on Mature Bacterial Biofilm

The inhibitory activity of the EOs on the mature biofilms was assessed using the tests described above, on bacterial cultures grown for 24 h. After 24 h of bacterial growth, the planktonic cells were removed, and the two concentrations of the EOs, 10 and 20 μL/mL, and Luria–Bertani broth were added, and a final volume of 250 μL/well was reached. Plates were kept for 24 h at 37 °C or 35 °C, depending on the strains. The other experimental steps, and the calculation of the percent value of inhibition with respect to the untreated bacteria, were performed as previously described.

#### 3.5.5. Effects of the EOs on Cell Metabolic Activity within the Biofilm

The 3-(4,5-dimethylthiazol-2-yl)-2,5-diphenyltetrazolium bromide (MTT) colorimetric method [42] was used to evaluate the effect of the EOs on the metabolic activity of the bacterial cells. Two concentrations (10 and 20 μL/mL) of the EOs, which were added at the beginning of the bacterial growth and after 24 h of incubation, were used. After 48 h total incubation, planktonic cells were removed; addition of 150 μL of PBS and 30 μL of 0.3% MTT (Sigma, Milano, Italy) was followed by the incubation of the microplates for two hours at 37 °C (35 °C for *A. baumannii*). The MTT solution was removed, two washing steps were carried out with 200 μL of sterile physiological solution, and 200 μL of dimethyl sulfoxide (DMSO) was added to permit the suspension of the formazan crystals, assessed after two hours at λ = 570 nm (Cary Varian, Milano, Italy).

### 3.6. Statistical Analysis

All experiments were carried out in triplicate. Data from each experiment were statistically analyzed using GraphPad Prism 6.0 software (GraphPad Software Inc., San Diego, CA, USA) followed by comparison of means (two-way ANOVA) using Dunnett’s multiple comparisons test, at the significance level of *p* < 0.05.

## 4. Conclusions

This research has allowed the chemical study of the EOs of seven species of *Eucalyptus* grown outside their area of origin and to help in the study of the chemodiversity of a complex genus such as the *Eucalyptus* genus.

The results evidenced the phytotoxic activity of the seven *Eucalyptus* species, already reported for other *Eucalyptus* species. Moreover, all EOs could be considered as inhibitory to the bacterial adhesion and the cellular metabolism of all the pathogenic strains used in our experiments. This efficacy was also observed in tests performed on mature biofilms. Considering these results, further investigations on these EOs, such as the molecular analysis of the genes involved in the various steps of adhesion and mature biofilm formation, as well as the assessment of toxicity of the EOs, could provide useful in applications for both the agronomic management and the treatment of human microbial infections.

## Figures and Tables

**Figure 1 molecules-27-08227-f001:**
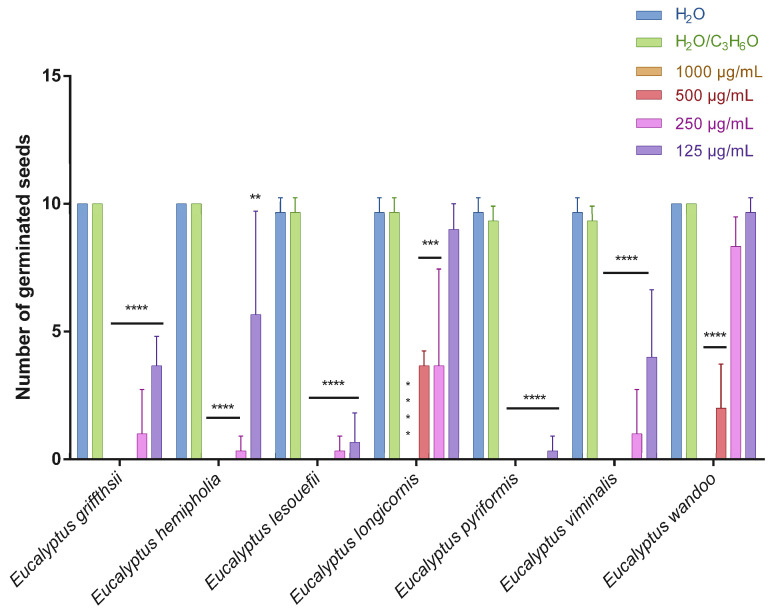
Number of germinated seeds of *Sinapis arvensis* seeds after treatment with different concentrations (125–1000 μg/mL) of *Eucalyptus* essential oils measured 120 h after sowing. Results are the mean ± standard deviation (*n* = 3) of three independent experiments. * *p* < 0.05; ** *p* < 0.01, *** *p* < 0.001, **** *p* < 0.0001 compared to control (ANOVA followed by Dunnett’s multiple comparison test).

**Figure 2 molecules-27-08227-f002:**
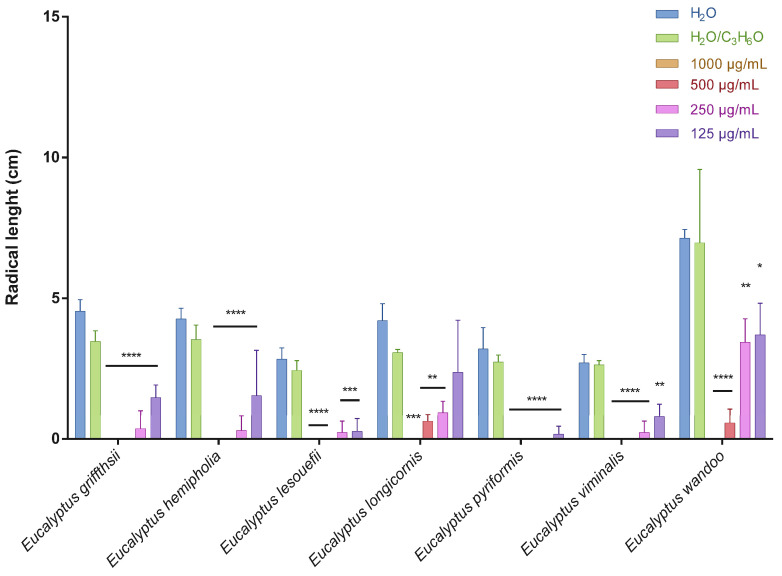
Radical length of *Sinapis arvensis* seeds after treatment with different concentrations (125–1000 μg/mL) of *Eucalyptus* essential oils measured 120 h after sowing. Results are the mean ± standard deviation (*n* = 3) of three independent experiments. * *p* < 0.05; ** *p* < 0.01, *** *p* < 0.001, **** *p* < 0.0001 compared to control (ANOVA followed by Dunnett’s multiple comparison test).

**Figure 3 molecules-27-08227-f003:**
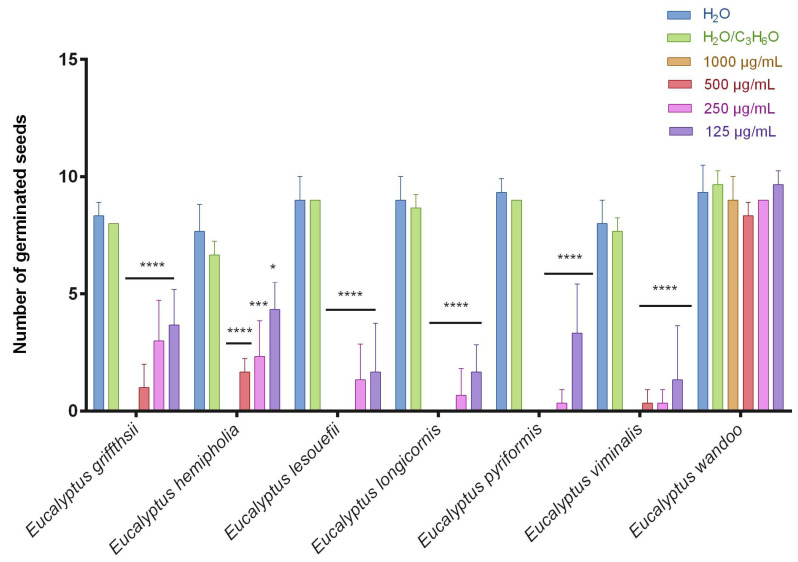
Number of germinated seeds of *Raphanus sativus* seeds after treatment with different concentrations (125–1000 μg/mL) of *Eucalyptus* essential oils measured 120 h after sowing. Results are the mean ± standard deviation (*n* = 3) of three independent experiments. * *p* < 0.05; *** *p* < 0.001, **** *p* < 0.0001 compared to control (ANOVA followed by Dunnett’s multiple comparison test).

**Figure 4 molecules-27-08227-f004:**
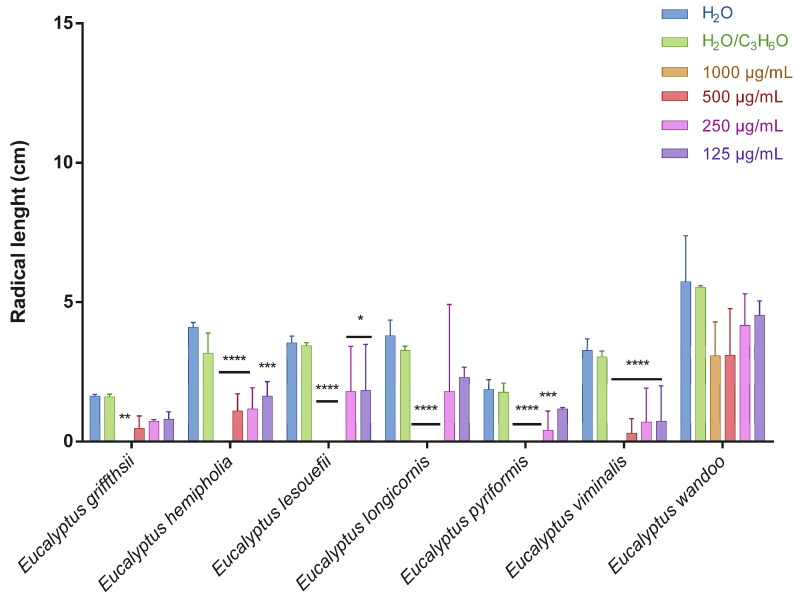
Radical length of *Raphanus sativus* seeds after treatment with different concentrations (125–1000 μg/mL) of *Eucalyptus* essential oils measured 120 h after sowing. Results are the mean ± standard deviation (*n* = 3) of three independent experiments. * *p* < 0.05; ** *p* < 0.01, *** *p* < 0.001, **** *p* < 0.0001 compared to control (ANOVA followed by Dunnett’s multiple comparison test).

**Figure 5 molecules-27-08227-f005:**
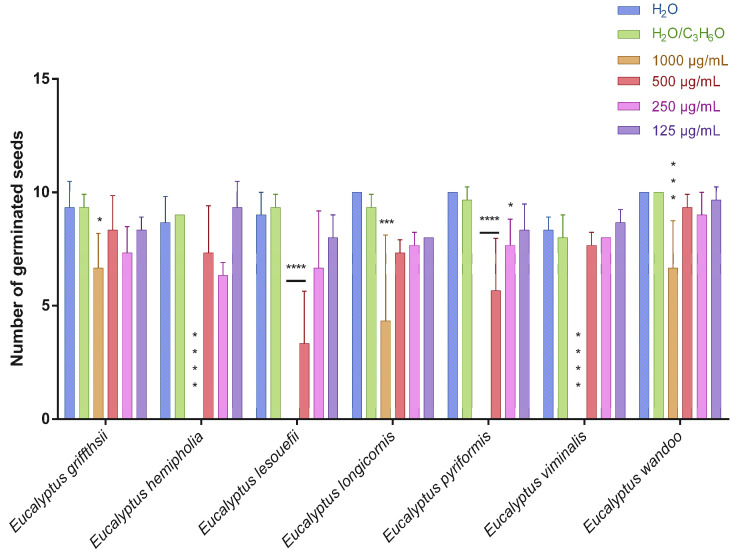
Number of germinated seeds of *Lolium multiflorum* seeds after treatment with different concentrations (125–1000 μg/mL) of *Eucalyptus* essential oils measured 120 h after sowing. Results are the mean ± standard deviation (*n* = 3) of three independent experiments. * *p* < 0.05; *** *p* < 0.001, **** *p* < 0.0001 compared to control (ANOVA followed by Dunnett’s multiple comparison test).

**Figure 6 molecules-27-08227-f006:**
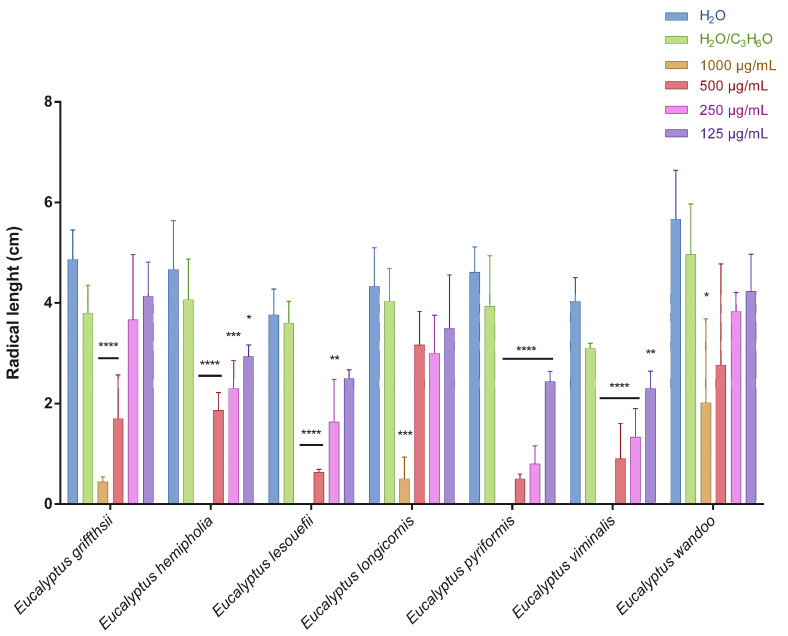
Radical length of *Lolium multiflorum* seeds after treatment with different concentrations (125–1000 μg/mL) of *Eucalyptus* essential oils measured 120 h after sowing. Results are the mean ± standard deviation (*n* = 3) of three independent experiments. * *p* < 0.05; ** *p* < 0.01, *** *p* < 0.001, **** *p* < 0.0001 compared to control (ANOVA followed by Dunnett’s multiple comparison test).

**Table 1 molecules-27-08227-t001:** Essential Oil Yields of seven *Eucalyptus* Species.

	Yield (%)
*Eucalyptus griffithsii*	1.8
*Eucalyptus hemiphloia*	3.6
*Eucalyptus lesouefii*	3.2
*Eucalyptus longicornis*	1.7
*Eucalyptus pyriformis*	2.8
*Eucalyptus viminalis*	1.8
*Eucalyptus wandoo*	1.3

**Table 2 molecules-27-08227-t002:** Chemical composition of the EOs [*E. griffithsii* (EG), *E. hemiphloia* (EH), *E. lesoufii* (ELE), *E. longicornis* (ELO), *E. pyriformis* (EP), *E. viminalis* (EV), *E. wandoo* (EW)+].

N	Compound Name	EG	EH	ELE	ELO	EP	EV	EW	KI ^a^	KI ^b^	Identification ^c^
1	α-Thujene	0.1	-	-	-	-	-	-	859	930	1, 2, 3
2	α-Pinene	9.8	0.6	14.6	3.5	1.9	7.2	4.3	864	939	1, 2, 3
3	Camphene	0.2	-	0.2	0.2	0.1	0.1	0.1	876	954	1, 2, 3
4	Thuja-2,4(10)-diene	-	-	-	-	t	-	-	882	960	1, 2
5	β-Pinene	3.8	0.1	4.4	0.1	3.5	0.1	-	902	979	1, 2, 3
6	Myrcene	-	0.1	-	-	-	-	-	923	990	1, 2, 3
7	δ-3-Carene	0.1	-	-	-	-	-	-	930	1002	1, 2, 3
8	α-Phellandrene	-	0.5	0.2	-	0.3	0.1	-	931	1011	1, 2, 3
9	α-Terpinene	0.1	0.3	-	-	0.4	-	-	944	1017	1, 2, 3
10	*o*-Cymene	4.2	-	-	-	-	-	-	951	1024	1, 2, 3
11	*p*-Cymene	-	5.5	14.3	-	28.8	-	0.6	952	1026	1, 2, 3
12	Eucalyptol	48.9	63.2	40.8	84.2	15.1	68.1	73.6	958	1031	1, 2, 3
13	(Z)-β-Ocimene	-	-	-	-	0.4	-	-	983	1037	1, 2, 3
14	γ-Terpinene	0.6	1.3	0.5		-	-	1.1	984	1059	1, 2, 3
15	*cis*-Sabinene hydrate	-	-	-	-	0.1	-	-	994	1070	1, 2
16	*cis*-Linalool oxide	-	0.3	-	-	0.2	-	-	997	1072	1, 2
17	Terpinolene	0.1	-	-	-	-	-	-	1007	1088	1, 2, 3
18	*p*-Mentha-3,8-diene	-	-	-	-	0.3	-	-	1008	1072	1, 2
19	*trans*-Linalool oxide	-	0.3	-	-	-	-	-	1009	1086	1, 2
20	*m*-Cymenene	-	-	-	-	0.5	-	-	1010	1085	1, 2
21	*p*-Cymenene	0.2	0.3	-	-	-	-	-	1010	1091	1, 2, 3
22	α-Pinene oxide	-	-	0.1	-	-	-	-	1010	1099	1, 2
23	6-Camphenol	-	-	-	-	-	0.2	-	1010	1113	1, 2
24	6-Camphenone	-	-	-	-	0.1	-	0.1	1018	1096	1, 2
25	Linalool	-	0.4	-	-	-	-	-	1024	1096	1, 2, 3
26	*cis*-Thujone	0.4	0.2	-	-	0.4	-	-	1022	1102	1, 2, 3
27	Isoamyl isovalerate	0.6	0.5	0.4	0.4	-	-	-	1029		1, 2
28	Isopentyl isovalerate	-	-	-	-	0.1	-	-	1029		1, 2
29	*exo*-Fenchol	-	0.1	0.6	0.4	-	0.4	0.2	1031	1121	1, 2, 3
30	*trans*-Thujol	0.1	-	-	-	0.4	-	-	1033		1, 2, 3
31	*trans*-*p*-Mentha-2,8-dien-1-ol	-	3.2	-	-	-	-	-	1040	1122	1, 2
32	*cis*-*p* -Menth-2-en-1-ol	0.1	-	-	-	0.3	-	-	1041	1121	1, 2
33	α-Campholenal	0.4	0.4	0.3	0.1	0.8	0.2	0,1	1043	1126	1, 2
34	cis- *p*-Mentha-2,8-dien-1-ol	-	-	-	-	0.1	-	-	1047	1137	1, 2
35	*allo*-Ocimene	0.3	-	-	-	-	0.1	-	1051	1132	1, 2, 3
36	Terpineol	-	-	0.4	-	-	-	-	1051	1133	1, 2, 3
37	*iso*-3-Thujanol	-	-	-	-	0.7	-	-	1052	1138	1, 2
38	*trans*-Pinocarveol	4.2	0.3	4.9	4.2	4.4	3.9	6.1	1055	1139	1, 2
39	Camphor	0.2	-	-	0.4	-	-	-	1058	1146	1, 2, 3
40	*trans*-*p*-Menth-2-en-1-ol	-	0.3	-	-	0.3	-	-	1059	1140	1, 2
41	Sabina ketone	2.0	-	1.4	-	3.9	1.2	-	1067	1159	1, 2, 3
43	*cis*-Verbenol	0.3	-	-	-	0.2	-	-	1064	1141	1, 2, 3
44	Pinocarvone	1.3	0.3	1.3	0.9	1.3	1.2	1.3	1077	1164	1, 2, 3
45	Borneol	0.6	0.4	0.9	0.5	0.3	0.5	0.4	1084	1169	1, 2, 3
46	Menthol	-	-	-	-	-	0.2	-	1084	1171	1, 2, 3
47	*p*-Mentha-1,5-dien-8-ol	-	-	-	0.2	0.6	-	-	1084	1170	1, 2
48	Umbellunone	0.3	-	-	-	-	-	-	1087	1171	1, 2
49	*trans*-2-hydroxy-Pinocamphone	-	-	-	-	-	-	0.7	1093	1250	1, 2
50	Terpinen-4-ol	1.8	1.7	1.5	0.1	3	1.1	-	1094	1177	1, 2
51	*neo*-Verbenol	2.9	-	1.6	-	-	-	-	1093		1, 2
52	Cryptone	-	-	-	-	-	1.4	-	1094	1185	1, 2
53	*m*-Cymen-8-ol	-	-	-	-	8.8	-	-	1096	1179	1, 2
54	*iso*-Menthol	-	4.6	-	-	-	-	-	1096	1182	1, 2
55	*p*-Cymen-8-ol	0.6	1.0	-	-	-	-	-	1099	1182	1, 2, 3
56	*cis*-Pinocarveol	-	-	-	0.4	-	-	-	1098	1184	1, 2
58	α-Terpinyl acetate	-	-	-	-	-	-	1.9	1101	1349	1, 2
59	Dihydro carveol	1.6	-	3.9	-	-	0.2	0.1	1102	1193	1, 2
60	α-Terpineol	1.9	1.2	1.7	0.8	-	-	0.1	1102	1188	1, 2, 3
61	γ-Terpineol	-	-	-	-	1.9	1.0	-	1104	1199	1, 2, 3
62	*trans*-Carveol	-	-	-	-	1.7	-	-	1109	1216	1, 2
63	Myrtenol	0.2	-	-	-	-	-	-	1112	1195	1, 2, 3
64	*trans*-Pulegol	-	-	-	-	-	0.3	-	1112	1214	1, 2, 3
65	Verbenone	0.3	0.8	-	-	1.7	-	-	1113	1205	1, 2, 3
66	4-Methyleneisophorone	-	-	-	-	0.7	-	-	1118		1, 2
67	*cis*-Carveol	0.2	-	-	-	-	-	-	1132	1229	1, 2, 3
68	*cis*-*p*-Mentha-1(7),8-dien-2-ol	-	-	-	-	0.3	-	-	1133	1230	1, 2
69	(E)-Ocimenone	0.3	-	-	-	-	-	-	1141	1238	1, 2
70	(Z)-Ocimenone	-	0.3	-	-	-	-	-	1142	1229	1, 2
71	Pulegone	-	-	-	-	0.1	-	-	1142	1237	1, 2, 3
72	Cumin aldehyde	1.9	4	1.0	-	5.2	1.0	-	1150	1241	1, 2, 3
73	Carvone	0.9	-	-	-	-	-	-	1155	1243	1, 2, 3
74	*trans*-*p*-Mentha-1(7),8-dien-2-ol	-	-	-	0.5	-	0.7	-	1143	1189	1, 2
75	*trans*-Piperitol acetate	-	-	-	-	2.2	-	-	1154	1346	1, 2
76	Piperitone	0.4	0.6	-	-	-	-	-	1163	1252	1, 2
77	*cis*-p-Mentha-8-thiol-3-one	-	-	-	-	2.4	-	-	1182	1360	1, 2
78	Perilla aldehyde	-	2.2	-	-	-	-	-	1182	1271	1, 2, 3
79	*cis*-Carvone oxide	-	-	-	-	-	0.3	-	1183	1263	1, 2
80	*p*-Menth-1-en-7-al	1.2	-	0.5	-	-	0.2	-	1184	1275	1, 2
81	α-Terpinen-7-al	0.2	0.3	-	-	-	-	-	1198	1285	1, 2
82	*p*-Cymen-7-ol	0.6	-	-	-	-	-	-	1204	1290	1, 2
83	Carvacrol acetate	-	-	-	-	1.9	-	0.1	1208	1372	1, 2
84	Carvacrol	1.0	1.8	-	-	-	0.3	-	1230	1299	1, 2, 3
85	δ-Elemene	-	-	0.1	-	-	-	-	1230	1338	1, 2, 3
86	Piperitenone	0.3	-	-	-	-	-	-	1244	1343	1, 2
87	*trans*-Carvyl acetate	-	-	-	0.1	-	-	0.6	1244	1342	1, 2
88	(E)-Jasmonyl acetate	-	-	-	-	0.5	-	-	1252	1398	1, 2
89	*allo*-Aromadendrene-	-	-	-	-	-	0.6	-	1327	1460	1, 2, 3
90	dehydro-Aromadendrane	-	-	-	-	-	0.4	-	1348	1462	1, 2
91	α-Vetispirene	-	-	-	-	-	0.1	-	1376	1490	1, 2
92	*cis*-β-Guaiene	-	-	-	-	0.2	-	-	1384	1493	1, 2
93	Viridiflorene	-	-	-	-	-	0.1	0.6	1383	1496	1, 2, 3
94	2-Propyl-heptanol	-	-	-	0.1	-	-	-	1389		1, 2
95	Phenetyl pivalate	-	-	-	0.3	-	-	-	1393		1, 2
96	Germacrene B	-	-	-	-	-	-	0.3	1441	1561	1, 2, 3
97	Spathulenol	1.3	0.2	1.2	-	0.2	5.1	2.1	1459	1578	1, 2, 3
98	Caryophyllene oxide	-	0.2	-	-	0.4	-	-	1462	1583	1, 2, 3
99	Guaiol	0.2	-	-	-	-	-	-	1464	1600	1, 2, 3
100	Viridiflorol	-	-	-	-	0.3	-	0.2	1465	1592	1, 2, 3
101	Globulol	-	-	-	0.1	-	0.2	-	1466	1590	1, 2, 3
102	Cubeban-11-ol	-	-	-	-	-	-	0.1	1474	1595	1, 2, 3
103	Rosifoliol	-	-	-	-	-	0.1	1.0	1496	1600	1, 2, 3
104	1-*epi*-Cubenol	-	-	-	-	0.1	-	-	1502	1628	1, 2, 3
105	α-Eudesmol	-	-	-	-	0.2	-	0.9	1524	1653	1, 2, 3
106	β-Eudesmol	-	-	-	0.1	-	-	1.5	1526	1650	1, 2, 3
	**Total**	96.7	97.5	96.8	97.6	97.3	96.6	98.0			
	Monoterpene hydrocarbons	19.5	9.1	34.2	3.8	36.2	7.6	6.1			
	Oxygenated monoterpenes	73.2	81.3	59.9	92.8	53.9	81.4	85.2			
	Sesquiterpene hydrocarbons	-	-	0.1	-	0.7	1.2	0.9			
	Oxygenated sesquiterpenes	1.5	0.4	1.2	0.2	1.2	5.4	5.8			
	Other	2.5	6.7	1.4	0.8	5.3	1	-			

^a^ Kovats index determined relative to the tR of a series of n-alkanes (C_10_-C_35_) on an HP-5MS column; ^b^ Kovats index from literature [23]; ^c^ Identification method: 1 = linear retention index; 2 = identification based on the comparison of mass spectra; 3 = Co-injection with standard compounds; t = traces, less than 0.1%; - = absent.

**Table 3 molecules-27-08227-t003:** MIC (µL/mL) of the EOs required to inhibit the growth of *A. baumannii*, *E. coli*, *L. monocytogenes, P. aeruginosa* and *S. aureus*. Tetracycline was used as a positive control.

MIC (µL/mL)					
	*A. baumannii*	*E. coli*	*L. monocytogenes*	*P. aeruginosa*	*S. aureus*
*E. griffithsii*	25 ± 1 ^**^	25 ± 1	26 ± 1	26 ± 1	25 ± 1
*E. hemiphloia*	26 ± 2	26 ± 1 ^*^	28 ± 1 ^**^	30 ± 2 ^**^	28 ± 1 ^***^
*E. lesoufii*	28 ± 1 ^**^	28 ± 1 ^***^	26 ± 1	27 ± 2	27 ± 1 ^**^
*E. longicornis*	30 ± 1 ^***^	30 ± 1 ^****^	38 ± 2 ^****^	27 ± 2	35 ± 1 ^****^
*E. pyriformis*	25 ± 1	25 ± 1	26 ± 2	26 ± 1	26 ± 2 ^*^
*E. viminalis*	30 ± 1 ^***^	28 ± 1 ^***^	25 ± 1	25 ± 1	30 ± 1 ^****^
*E. wandoo*	26 ± 1	29 ± 1 ^****^	35 ± 1 ^****^	26 ± 1	29 ± 1 ^****^
Tetracycline	24 ± 2	23 ± 1	23 ± 1	24 ± 2	23 ± 1

Data are expressed as the mean ± SD of three experiments; * *p* < 0.05; ** *p* < 0.01, *** *p* < 0.001, **** *p* < 0.0001 compared to control (ANOVA followed by Dunnett’s multiple comparison test).

**Table 4 molecules-27-08227-t004:** Inhibitory activity of the EOs on the biofilm, at 0 and 24 h.

Time 0	*A. baumannii*	*E. coli*	*L. monocytogenes*	*P. aeruginosa*	*S. aureus*
EG 10 µL/mL	82.48 ± 0.2 ^****^	85.44 ± 0.2 ^****^	85.03 ± 0.1 ^****^	89.62 ± 0.2 ^****^	87.17 ± 0.1 ^****^
EG 20 µL/mL	92.53 ± 0.1 ^****^	86.39 ± 0.2 ^****^	87.56 ± 0.2 ^****^	91.40 ± 0.1 ^****^	93.73 ± 0.1 ^****^
EH 10 µL/mL	37.53 ± 0.2 ^****^	47.00 ± 0.3 ^****^	77.59 ± 0.3 ^****^	78.81 ± 0.2 ^****^	84.23 ± 0.1 ^****^
EH 20 µL/mL	74.98 ± 0.3 ^****^	78.44 ± 0.4 ^****^	81.06 ± 0.3 ^****^	79.71 ± 0.3 ^****^	84.99 ± 0.2 ^****^
EP 10 µL/mL	92.08 ± 0.2 ^****^	89.71 ± 0.2 ^****^	79.13 ± 0.2 ^****^	84.81 ± 0.1 ^****^	88.51 ± 0.1 ^****^
EP 20 µL/mL	92.37 ± 0.1 ^****^	94.39 ± 0.1 ^****^	86.65 ± 0.2 ^****^	90.64 ± 0.1 ^****^	95.04 ± 0.2 ^****^
ELE 10 µL/mL	73.89 ± 0.3 ^****^	60.58 ± 0.3 ^****^	0.00 ± 0.00	64.08 ± 0.3 ^****^	77.32 ± 0.2 ^****^
ELE 20 µL/mL	79.61 ± 0.2 ^****^	89.92 ± 0.2 ^****^	92.14 ± 0.1 ^****^	89.48 ± 0.1 ^****^	86.03 ± 0.2 ^****^
ELO 10 µL/mL	84.82 ± 0.1 ^****^	85.64 ± 0.2 ^****^	82.19 ± 0.3 ^****^	80.91 ± 0.2 ^****^	62.11 ± 0.3 ^****^
ELO 20 µL/mL	86.09 ± 0.1 ^****^	88.16 ± 0.1 ^****^	88.14 ± 0.3 ^****^	87.38 ± 0.2 ^****^	73.98 ± 0.2 ^****^
EV 10 µL/mL	59.39 ± 0.1 ^****^	54.29 ± 0.2 ^****^	0.00 ± 0.00	83.31 ± 0.3 ^****^	4.36 ± 0.1 ^****^
EV 20 µL/mL	63.16 ± 0.2 ^****^	77.07 ± 0.2 ^****^	98.67 ± 0.1 ^****^	93.70 ± 0.1 ^****^	58.84 ± 0.2 ^****^
EW 10 µL/mL	89.15 ± 0.2 ^****^	77.64 ± 0.1 ^****^	0.00 ± 0.00	89.24 ± 0.2 ^****^	59.16 ± 0.2 ^****^
EW 20 µL/mL	89.43 ± 0.2 ^****^	91.00 ± 0.1 ^****^	88.78 ± 0.2 ^****^	90.09 ± 0.1 ^****^	75.90 ± 0.2 ^****^
**24 h**	** *A. baumannii* **	** *E. coli* **	** *L. monocytogenes* **	** *P. aeruginosa* **	** *S. aureus* **
EG 10 µL/mL	67.88 ± 0.3 ^****^	33.66 ± 0.2 ^****^	24.31 ± 0.2 ^****^	78.64 ± 0.2 ^****^	71.21 ± 0.4 ^****^
EG 20 µL/mL	68.39 ± 0.2 ^****^	33.31 ± 0.2 ^****^	35.76 ± 0.1 ^****^	81.79 ± 0.3 ^****^	87.15 ± 0.2 ^****^
EH 10 µL/mL	9.32 ± 0.1 ^****^	52.23 ± 0.2 ^****^	6.10 ± 0.1 ^****^	53.15 ± 0.2 ^****^	73.93 ± 0.2 ^****^
EH 20 µL/mL	51.64 ± 0.2 ^****^	53.28 ± 0.2 ^****^	25.63 ± 0.2 ^****^	59.50 ± 0.1 ^****^	75.65 ± 0.3 ^****^
EP 10 µL/mL	52.16 ± 0.2 ^****^	26.85 ± 0.2 ^****^	6.18 ± 0.1 ^****^	59.35 ± 0.2 ^****^	82.33 ± 0.2 ^****^
EP 20 µL/mL	71.31 ± 0.2 ^****^	36.34 ± 0.2 ^****^	67.88 ± 0.2 ^****^	87.56 ± 0.3 ^****^	88.84 ± 0.3 ^****^
ELE 10 µL/mL	52.15 ± 0.2 ^****^	62.56 ± 0.2 ^****^	0.00 ± 0.00	34.58 ± 0.2 ^****^	52.13 ± 0.4 ^****^
ELE 20 µL/mL	70.90 ± 0.3	65.96 ± 0.2 ^****^	61.19 ± 0.2 ^****^	57.35 ± 0.2 ^****^	58.20 ± 0.4 ^****^
ELO 10 µL/mL	62.15 ± 0.3 ^****^	1.77 ± 0.2 ^*^	63.33 ± 0.2 ^****^	52.63 ± 0.3 ^****^	55.23 ± 0.3 ^****^
ELO 20 µL/mL	68.94 ± 0.2 ^****^	50.42 ± 0.2 ^****^	65.22 ± 0.1 ^****^	52.71 ± 0.2 ^****^	60.41 ± 0.3 ^****^
EV 10 µL/mL	14.70 ± 0.1 ^****^	0.30 ± 0.2	0.00 ± 0.00	5.49 ± 0.1 ^***^	50.77 ± 0.2 ^****^
EV 20 µL/mL	25.29 ± 0.1 ^****^	39.64 ± 0.2 ^****^	4.39 ± 0.1 ^****^	20.44 ± 0.1 ^****^	53.49 ± 0.1 ^****^
EW 10 µL/mL	30.58 ± 0.1 ^****^	25.46 ± 0.2 ^****^	0.00 ± 0.00	56.68 ± 0.2 ^****^	94.55 ± 0.1 ^****^
EW 20 µL/mL	36.92 ± 0.2 ^****^	41.65 ± 0.2 ^****^	57.27 ± 0.2 ^****^	63.53 ± 0.1 ^****^	95.83 ± 0.1 ^****^

The data are the inhibition percentages calculated with respect to the untreated bacteria used as control and are reported as the mean of three independent experiments ± SD, * *p* < 0.05, ****p* < 0.001,. *****p* < 0.0001 compared with control (ANOVA followed by Dunnett’s multiple comparison test). EG = *E. griffithsii*, EH = *E. hemiphloia*, ELE = *E. lesoufii*, ELO = *E. longicornis*, EP = *E. pyriformis*, EV = *E. viminalis*, EW = *E. wandoo*.

**Table 5 molecules-27-08227-t005:** Inhibitory activity of the EOs on the metabolism of the bacterial sessile cells in the immature biofilm, at 0 and 24 h.

Time 0	*A. baumannii*	*E. coli*	*L. monocytogenes*	*p. aeruginosa*	*S. aureus*
EG 10 µL/mL	83.35 ± 0.3 ^****^	75.29 ± 0.3 ^****^	80.04 ± 0.2 ^****^	79.71 ± 0.3 ^****^	79.03 ± 0.2 ^****^
EG 20 µL/mL	85.12 ± 0.3 ^****^	78.23 ± 0.3 ^****^	83.22 ± 0.1 ^****^	81.93 ± 0.2 ^****^	81.77 ± 0.1 ^****^
EH 10 µL/mL	82.30 ± 0.1 ^****^	69.79 ± 0.3 ^****^	79.21 ± 0.2 ^****^	75.79 ± 0.1 ^****^	79.64 ± 0.3 ^****^
EH 20 µL/mL	83.74 ± 0.1 ^****^	79.41 ± 0.3 ^****^	79.48 ± 0.3 ^****^	78.77 ± 0.2 ^****^	80.70 ± 0.2 ^****^
EP 10 µL/mL	80.37 ± 0.1 ^****^	48.89 ± 0.1 ^****^	39.63 ± 0.1 ^****^	72.40 ± 0.2 ^****^	29.93 ± 0.2 ^****^
EP 20 µL/mL	83.98 ± 0.2 ^****^	79.52 ± 0.3 ^****^	73.00 ± 0.2 ^****^	78.49 ± 0.3 ^****^	80.94 ± 0.1 ^****^
ELE 10 µL/mL	0.00 ± 0.00	0.00 ± 0.00	0.00 ± 0.00	0.00 ± 0.00	0.00 ± 0.00
ELE 20 µL/mL	71.98 ± 0.3 ^****^	90.71 ± 0.1 ^****^	82.25 ± 0.1 ^****^	74.74 ± 0.2 ^****^	88.08 ± 0.3 ^****^
ELO 10 µL/mL	21.26 ± 0.2 ^****^	72.66 ± 0.1 ^****^	0.00 ± 0.00	43.38 ± 0.4 ^****^	0.27 ± 0.00
ELO 20 µL/mL	30.43 ± 0.2 ^****^	74.94 ± 0.2 ^****^	15.59 ± 0.1 ^****^	53.92 ± 0.3 ^****^	41.34 ± 0.3 ^****^
EV 10 µL/mL	22.36 ± 0.1 ^****^	58.87 ± 0.2 ^****^	78.17 ± 0.3 ^****^	65.27 ± 0.3 ^****^	70.23 ± 0.2 ^****^
EV 20 µL/mL	53.64 ± 0.2 ^****^	70.03 ± 0.1 ^****^	92.76 ± 0.1 ^****^	65.13 ± 0.3 ^****^	79.60 ± 0.4 ^****^
EW 10 µL/mL	52.84 ± 0.2	20.14 ± 0.1	20.67 ± 0.1 ^****^	1.06 ± 0.00 ^****^	11.91 ± 0.1
EW 20 µL/mL	57.60 ± 0.4 ^****^	63.66 ± 0.2	75.41 ± 0.4 ^****^	74.42 ± 0.2 ^****^	55.43 ± 0.2 ^****^
**24 h**	** *A. baumannii* **	** *E. coli* **	** *L. monocytogenes* **	** *P. aeruginosa* **	** *S. aureus* **
EG 10 µL/mL	0.00 ± 0.00	20.59 ± 0.2 ^****^	0.00 ± 0.00	0.00 ± 0.00	7.95 ± 0.2 ^****^
EG 20 µL/mL	0.00 ± 0.00	58.68 ± 0.2 ^****^	0.00 ± 0.00	0.00 ± 0.00	28.29 ± 0.2 ^****^
EH 10 µL/mL	0.00 ± 0.00	31.89 ± 0.2 ^****^	0.00 ± 0.00	0.00 ± 0.00	16.72 ± 0.2 ^****^
EH 20 µL/mL	0.00 ± 0.00	36.54 ± 0.2 ^****^	0.00 ± 0.00	0.00 ± 0.00	21.38 ± 0.2 ^****^
EP 10 µL/mL	2.09 ± 0.00 ^****^	33.52 ± 0.2 ^****^	29.90 ± 0.2 ^****^	0 ± 0.2	30.88 ± 0.2 ^****^
EP 20 µL/mL	35.07 ± 0.2 ^****^	51.34 ± 0.2 ^****^	54.04 ± 0.2 ^****^	7.09 ± 0.2 ^****^	33.22 ± 0.2 ^****^
ELE 10 µL/mL	0.00 ± 0.00	31.89 ± 0.2 ^****^	0.00 ± 0.00	33.95 ± 0.2	11.46 ± 0.2 ^****^
ELE 20 µL/mL	86.99 ± 0.1 ^****^	34.32 ± 0.2 ^****^	0.00 ± 0.00	0.00 ± 0.00	18.77 ± 0.2 ^****^
ELO 10 µL/mL	0.00 ± 0.00	16.48 ± 0.2 ^****^	0.00 ± 0.00	4.16 ± 0.00 ^****^	8.67 ± 0.2 ^****^
ELO 20 µL/mL	0.00 ± 0.00	38.92 ± 0.2 ^****^	0.00 ± 0.00	17.08 ± 0.2 ^****^	18.97 ± 0.2 ^****^
EV 10 µL/mL	0.00 ± 0.00	0 ± 0.2 ^****^	27.16 ± 0.2 ^****^	42.96 ± 0.2 ^****^	21.21 ± 0.2 ^****^
EV 20 µL/mL	65.38 ± 0.3 ^****^	38.39 ± 0.2 ^****^	27.90 ± 0.2 ^****^	83.43 ± 0.1 ^****^	73.94 ± 0.2 ^****^
EW 10 µL/mL	0.00 ± 0.00	10.32 ± 0.2 ^****^	0.00 ± 0.00	0.00 ± 0.00	18.43 ± 0.2 ^****^
EW 20 µL/mL	19.87 ± 0.2 ^****^	27.08 ± 0.2 ^****^	0.00 ± 0.00	9.73 ± 0.1 ^****^	30.54 ± 0.2 ^****^

The data are the inhibition percentages calculated with respect to the untreated bacteria used as control and are reported as the mean of three independent experiments ± SD, *****p* < 0.0001 compared with control (ANOVA followed by Dunnett’s multiple comparison test). EG = *E. griffithsii*, EH = *E. hemiphloia*, ELE = *E. lesoufii*, ELO = *E. longicornis*, EP = *E. pyriformis*, EV = *E. viminalis*, EW = *E. wandoo*.

**Table 6 molecules-27-08227-t006:** Date and place of harvest of samples.

	Date of Harvest	Srboretum (Region)
*Eucalyptus griffithsii*	April 2021	Henchir Naam (Siliana)
*Eucalyptus hemiphloia*	May 2021	Djebel Manasour (Zaghouen)
*Eucalyptus lesouefii*	April 2021	Henchir Naam (Siliana)
*Eucalyptus longicornis*	April 2021	Henchir Naam (Siliana)
*Eucalyptus pyriformis*	May 2021	Henchir Naam (Siliana)
*Eucalyptus viminalis*	May 2021	Souiniet (Jendouba)
*Eucalyptus wandoo*	May 2021	Djebel Manasour (Zaghouen)

## Data Availability

Not applicable.

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
