# Peer review of "Chemical Composition, Phytotoxic and Antibiofilm Activity of Seven Eucalyptus Species from Tunisia"

_molecules, 2022, doi:10.3390/molecules27238227_

Round 1

Reviewer 1 Report

In this article, the authors describe the chemical composition of 7 extracts of different types of eucalyptus collected in Tunisia, some of them being described for the first time in the literature.

With these extracts they study the herbicidal activity against 3 different weed seeds, the antimicrobial activity against gram + and gram - pathogens and finally the antibiofilm activity.

The work presented here appears to be an extension of a recent paper in Molecules (2022) in which other different eucalyptus extracts were studied and similar activity studies were performed.

The authors have been working on this subject for more than ten years, in relation to the papers they have published, so it is understood that the method for the determination of the chemical composition as well as the different experiments shown for the biological activities are well implemented.

The results are promising, especially those obtained in the antibiofilm studies.

The work is well written, maybe putting all the diagrams or figures together and then the text or vice versa, forces to go backwards or forwards looking for the information.

The graphs and tables are well understood and, finally, the conclusions are appropriate.

Author Response

The Authors thank the Referees for their comments which certainly contribute to improving the manuscript.
The manuscript was modified following all the suggestions of the Referees.

The point-by-point modifications are reported in the enclosed file

Reviewer 2 Report

Article entitled "Chemical composition, phytotoxic and antibiofilm activity of seven Eucalyptus species from Tunisia” is well written manuscript based on new elements in essential oils research, however manuscript has some points need to be improved. Below there is a list of suggested changes:   

-          line 52: the wording of the EO abbreviation should be unified in all text (EOs or Eos) 

-          line 19, 26, 27, 80…I proposes to write the entire name of the bacterial strain on one line. Do not separate one letter (E.). The same suggestion applies to compound names (line 94, 108…)

 -          line 122 – second word need to be corrected 

-          the spellings of “prefix” of the type of isomer like: iso, neo, Z, E, cis, trans – is usually written in italic letters (line 92, 

-          If the identification of EOs is based on GC-MS-RI (on semi-polar column) the experimental and literature data of RI of identified compounds should be add to the table 2. Otherwise how can we check the check the correctness of the analysis? 

-          Chapter 3.2: EO is product isolated during hydrodistillation or steam distillation (with an exception of citrus EOs), thus the title of chapter 3.2 should not be “Essential oil extraction”. 

-          Yield isolation and calculations - if the volume of the oil has been read with an accuracy to the first decimal place, the yield value should be given to the first decimal place. 

-          Table 2: 

o   the name of compounds 27, 94 and 95 look like copied directly from the Willey or NIST library. The name should be written according to IUPAC. 

o   compounds 16, 19 – oxide not Oxide o   mistake – compound no. 30. What it transT? 

o   compound 36 is Terpineol, not 1-terpineol. 

o   compound 66: 4-methyleneisophorone (name without dash) 

o   it is impossible to identified with capillary column (-) isomer of globulol (no. 101) 

o   some mistakes with the name of 104 compound   

In a conclusion despite the fact that the article is very well described, discussion is merit its key part needs to be improved

Author Response

(The authors gave the same response as above.)

Reviewer 3 Report

See attached file.

Most disturbing is the fact that the graphics are unreadable.

Author Response

(The authors gave the same response as above.)

Reviewer 4 Report

The manuscript molecules-2016992 “Chemical composition, phytotoxic and antibiofilm activity of seven Eucalyptus species from Tunisia” extracted the essential oils from seven Eucalyptus species, the chemical composition of essential oil was analyzed, and evaluated the phytotoxic and antibiofilm activity against several strains. I have made some comments on the manuscript.

1.       Line 27-28, some mistake in the sentence. Please rewrite it.

2.       The essential oil should be dried after being extracted by steam distillation. Please add how to dry the essential oil.

3.       The sterilization of essential oil method was not mentioned and maybe it could be added (heating, filtering, etc.).

4.       There are some problems with the evaluation of MICs. In fact, whether there is bacterial growth or not, the wells added the resazurin cannot become colorless. Authors should examine this section carefully and make corrections. The MIC should be “the lowest concentration of EOs which could prevent the solution from changing from dark purple to pink”.

5.       The MIC of the EOs to the tested strains is generally more than 20 μL. For the essential oil, this concentration is already high. Whether the author considers evaluating the toxicity of the EOs at this concentration, such as hemolysis test

6.       The authors should evaluate the effect of DOMO in the culture solution on the test strains.

7.       Please confirm that the detection wavelength of CV assay is 540 nm or 590 nm?

8.       Why did the authors not use XTT [2,3-bis (2-methyloxy 4-nitro-5-sulfophenyl)-2H-tetrazolium-5-carbox-anilide] to detect the cell metabolic activity within the biofilm, the sensitivity of XTT detection and the solubility of the formed complex are both better.

9.       The effects of EOs on the biofilms of the tested strains were analyzed by CV and MTT experiments. The results showed that EOs could inhibit biofilm. However, it is too early to directly conclude that EOs attenuates the adhesion of bacteria. It is suggested to increase the bacterial adhesion experiment or the detection of adhesion genes.

Author Response

(The authors gave the same response as above.)

Round 2

Reviewer 4 Report

It can be accepted in the present form.